# Development a Low-Cost Wireless Smart Meter with Power Quality Measurement for Smart Grid Applications

**DOI:** 10.3390/s23167210

**Published:** 2023-08-16

**Authors:** Ewerton L. de Sousa, Leonardo A. de Aquino Marques, Israel da S. Felix de Lima, Ana Beatriz M. Neves, Eduardo Nogueira Cunha, Márcio Eduardo Kreutz, Augusto José V. Neto

**Affiliations:** Department of Informatics and Applied Mathematics (DIMAp), Federal University of Rio Grande do Norte (UFRN), Natal 59078-970, RN, Brazil; leonardo.augusto.103@ufrn.edu.br (L.A.d.A.M.); israel.felix.075@ufrn.edu.br (I.d.S.F.d.L.); ana.neves.003@ufrn.edu.br (A.B.M.N.); eduardo.nogueira@ufrn.br (E.N.C.); kreutz@dimap.ufrn.br (M.E.K.); augusto@dimap.ufrn.br (A.J.V.N.)

**Keywords:** electricity, smart grid, smart meter, power quality measurement, ESP32, low cost

## Abstract

Developing a low-cost wireless energy meter with power quality measurements for smart grid applications represents a significant advance in efficient and accurate electric energy monitoring. In increasingly complex and interconnected electric systems, this device will be essential for a wide range of applications, such as smart grids, by introducing a real-time energy monitoring system. In light of this, smart meters can offer greater opportunities for sustainable and efficient energy use and improve the utilization of energy sources, especially those that are nonrenewable. According to the 2020 International Energy Agency (IEA) report, nonrenewable energy sources represent 65% of the global supply chain. The smart meter developed in this work is based on the ESP32 microcontroller and easily accessible components since it includes a user-friendly development platform that offers a cost-effective solution while ensuring reliable performance. The main objective of developing the smart meters was to enhance the software and simplify the hardware. Unlike traditional meters that calculate electrical parameters by means of complex circuits in hardware, this project performed the calculations directly on the microcontroller. This procedure reduced the complexity of the hardware by simplifying the meter design. Owing to the high-performance processing capability of the microcontroller, efficient and accurate calculations of electrical parameters could be achieved without the need for additional circuits. This software-driven approach with simplified hardware led to benefits, such as reduced production costs, lower energy consumption, and a meter with improved accuracy, as well as updates on flexibility. Furthermore, the integrated wireless connectivity in the microcontroller enables the collected data to be transmitted to remote monitoring systems for later analysis. The innovative feature of this smart meter lies in the fact that it has readily available components, along with the ESP32 chip, which results in a low-cost smart meter with performance that is comparable to other meters available on the market. Moreover, it is has the capacity to incorporate IoT and artificial intelligence applications. The developed smart meter is cost effective and energy efficient, and offers benefits with regard to flexibility, and thus represents an innovative, efficient, and versatile solution for smart grid applications.

## 1. Introduction

Since there is a constant growth in energy consumption in the world, continuous investment is required for expanding the energy matrix, and improving production and distribution processes.

In 1965, global energy consumption was 43 (TWh), but in 2022 it exceeded 165 TWh, an increase of almost 400% over the last five decades [1]. Two significant categories exist for the energy sources, comprising the world’s energy matrix: renewable and nonrenewable energy sources. Fossil fuels, such as oil, natural gas, and coal, are widely used throughout the world owing to their availability, reliability, and relatively low prices. However, these are nonrenewable energy sources and the primary sources of greenhouse gas emissions and air pollution.

The development of smart meters for SGs represents a solution to improve energy transmission between power plants and end consumers [2,3]. This transmission can be improved in various ways, such as real-time monitoring, quick detection and restoration of failures, as well as providing users with access to information about their consumption, and offering opportunities for savings. This approach aims to maximize energy efficiency, reduce waste, and meet the growing energy demand in a sustainable way.

In the current market, there is a very wide range of integrated specialized circuits for measuring electrical quantities. These integrated circuits offer several advantages when compared with conventional circuits, such as greater accuracy, smaller size, and lower power consumption.

However, specialized circuits tend to be more expensive owing to their complexity and precision, which may make using them unfeasible in low-cost projects. Furthermore, this type of circuit is less flexible than conventional circuits, especially in terms of customization or the capacity to be adapted to specific project requirements.

Another disadvantage is that it is not easy to integrate these circuits into a system, as they require more advanced technical knowledge and a deep understanding of their performance features, which may make it more difficult to implement these circuits in smaller-scale projects or projects that involve less experienced professionals.

Some examples of specialized integrated circuits for electrical measurements, such as the ADE9000, are highly accurate, fully integrated, multiphase energy and power quality monitoring devices. A superior analog performance and a DSP core provide accurate energy monitoring over a wide dynamic range. An integrated high-end reference ensures low drift when the temperature has a combined drift of less than ±25 (ppm/°C) maximum for the entire channel, including a PGA and an ADC. The integrated ADCs with a DSP engine calculate various parameters and provide data through user-accessible registers or indicate events through interrupt pins. With seven dedicated ADC channels, the ADE9000 can be used in a three-phase system or with up to three single-phase systems [4].

Another example is the MCP39F511, which is a highly integrated and full-featured single-phase power monitoring IC designed for the real-time measurement of input power for AC/DC power supplies, power distribution units, and industrial and consumer applications. This circuit includes dual-channel Delta-Sigma ADCs, a 16-bit calculation engine, EEPROM, and a flexible two-wire interface [5].

There is also the CS5490—a high-accuracy, two-channel, energy measurement analog front end, which incorporates independent 4th-order Delta-Sigma analog-to-digital converters for both channels, reference circuitry, and the proven EXL signal processing core, which allows it to provide active, reactive, and apparent energy measurement. In addition, RMS and power factor calculations are available. Calculations are energy output via a configurable energy pulse, or direct (UART) serial access to on-chip registers. Instantaneous current, voltage, and power measurements are also available through the serial port. The two-wire UART minimizes the cost of isolation where required [6].

These circuits feature multiple high-speed, high-precision ADCs and DSPs and control units to perform the necessary mathematical calculations and thus obtain key measurement parameters. Integrating all these components on a single chip makes the construction of these circuits complex and expensive.

As a result of the improvement of the microcontrollers, the chips are able to offer high processing power, with multi-processing cores, implement increasingly complex and advanced systems and thus ensure a considerable improvement in performance and energy efficiency. In addition, they offer a wide range of built-in features, such as communication peripherals, high-capacity memory, I/O device controllers, and even advanced security and encryption features. Owing to the increasing availability of these kinds of technologies on the market, it was possible to use these dedicated circuits for specialized software measurement circuits without loss of precision or quality, provide greater flexibility and reduce the final cost of the meter.

It is essential to construct low-cost smart meters to make this technology accessible to a broader audience. The reduction in production costs allows smart meters to be more widely adopted across different sectors and regions, as well as heightening an awareness of energy consumption, which leads to large-scale energy efficiency.

Furthermore, the greater popularity of smart meters encourages the transformation of electric grids into smart grids, thus meaning that renewable energy resources can be integrated with energy storage systems. The integration of renewable energy resources is achieved through real-time monitoring and energy power forecasting, and energy demand management, as well as better communications and real-time control, which lead to greater efficiency in the management and control of renewable sources of energy. This transition results in the grid being more flexible and resilient so that it can incorporate clean energy sources and make a shift towards a more sustainable energy matrix.

Smart grids are defined as an electrical network that can intelligently integrate the activities of all the connected users, such as generators and consumers, in order to provide energy efficiently, economically, and safely [7]. According to [8], the smart grid can be defined as the next-generation, intelligent electrical network that is capable of overcoming the most common failings of the existing grid. It is a network that intelligently integrates new technologies as a means of enhancing the monitoring and control operations of electrical systems, specifically with regard to power and distribution, while also incorporating the related measures taken by users. These networks are characterized by their ability to implement innovative devices and services, as well as new communication, control, monitoring, and self-diagnostic technologies within the system [9].

The operation of smart grids includes the use of software, hardware, and technologies that help electricity companies instantly detect and correct imbalances between generation and demand in order to improve the quality of service, increase the reliability of the energy system, and reduce costs [10]. It is precisely through these readings and the integration of smart grids that we find smart meters, which are an essential component of the smart grid. It is an electronic energy measurement device that replaces traditional meters, and provides significant benefits for the implementation and operation of an intelligent electrical network.

A smart meter is an advanced electronic device used to measure and monitor real-time electricity consumption. It differs from traditional meters by having additional features, such as the ability to communicate remotely and provide detailed information about energy consumption [11]. Smart meters have gained prominence worldwide because of the benefits they offer to both energy providers and consumers. These meters ensure more accurate and reliable measurements of energy consumption by dispensing with the need for manual readings and reducing reading errors [12]. Another important factor with regard to smart meters is their ability to understand power quality parameters.

Power quality is a crucial factor in energy measurement that should be taken into account, as it directly impacts the efficiency and safety of electrical systems. It involves various issues, such as distortions, interruptions, fluctuations, and other factors that affect the distribution of electrical energy. One of the key aspects of power quality is the occurrence of harmonics [13]. These harmonics refer to electrical currents or voltages with multiple frequencies in contrast with the fundamental frequency of the electric grid (usually 60 Hz). They arise from non-linear electrical equipment, such as motors, electronic lamps, frequency converters, and computers. The presence of harmonics can cause waveform distortions in voltage and current, resulting in energy losses, the overheating of equipment, and possible system failures [14].

Another critical aspect of power quality is the presence of voltage surges. Voltage surges are short peaks in voltage that can damage sensitive electronic equipment. They can be caused by lightning strikes, high-power equipment operations, or switching operations in the electric network. Monitoring and controlling the presence of voltage surges is essential for safeguarding electronic equipment. Voltage fluctuation is also a factor that affects power quality and occurs when the voltage in the electric network varies as a result of changes in the electric load. These kinds of occurrences can result in operational problems for electrical equipment, including motor failures and flickering lights. It is essential to maintain stable and controlled voltage to ensure power quality [13].

Assessing the quality of electrical energy is essential to ensure that electrical equipment and systems operate correctly and protect consumers from safety risks and damage to equipment [15,16]. Additionally, continuous monitoring of the quality of electrical energy can help electricity companies detect problems in the electric grid and take corrective measures to improve the efficiency and reliability of energy distribution [17]. This continuous monitoring is carried out by integrating IoT devices, such as smart meters and sensors.

The IoT is a new paradigm that is rapidly gaining popularity in today’s wireless landscape. The basic idea behind this concept is the widespread presence of various “things” or objects around us, such as RFID tags, sensors, actuators, cell phones, and other factors that can interact with each other and cooperate with their neighbors to attain common goals through unique addressing schemes [18]. It is clear that the IoT is increasing its applications and making cities safe and sustainable by having a greater impact on people’s lives [19].

Smart meters are intelligent electrical energy measurement systems that measure this consumption and transmit the collected data to a central server in real time [20]. By utilizing these data, stakeholders can make informed decisions about the electrical energy supply, and thus adjust it in accordance with the current demand, prevent consumption peaks, and reduce energy waste.

In light of these benefits, intelligent electricity metering is one of the most critical applications of the IoT, especially in the energy sector, as, in addition to monitoring energy consumption in real time, it can also find opportunities to save and reduce energy costs [21].

When making measurements, these devices acquire data from the electric network, such as voltage, current, and the power factor, to a central system through bidirectional communication [22]. Consumers can use the information acquired by the smart meter to improve their rate of consumption [17], while energy suppliers can monitor and bill the system [16].

In addition, smart meters make it easier to integrate variable renewable energy sources, such as solar and wind, and new loads, such as energy storage and electric vehicle charging, and thus ensure the stability and efficiency of the system [23]. These devices also allow consumers who produce their own energy to respond to market prices and sell the surplus to the grid. Feedback for consumption is essential, as smart meters can provide consumers with real-time feedback on energy consumption through devices such as smart home displays. This feedback may include comparisons with other consumers (normative feedback), which has the potential to encourage long-term energy conservation [23,24]. Studies show that normative feedback can encourage consumers to reduce consumption if they regard themselves as being above average or above their peers [25].

Thus, the use of traditional meters, such as the CW500, can be fully appreciated, although they remain useful mainly for specialized users. However, their applicability is becoming limited in an electrical system that is increasingly interconnected and volatile. This limitation stems from their high cost, usability, and the risks associated with handling.

Smart meters are designed to ensure that the quality of energy supplied is suitable for its intended use by end consumers or even the utilities that distribute electricity [16]. In addition to assessing whether the system is operating correctly, phenomena, random or intrinsic, occur in the electrical system, causing changes in the data reading. These phenomena include the following: sags and/or voltage swells, interruptions, harmonic distortions, voltage fluctuations, oscillatory or impulsive transients, noise, overvoltages, and undervoltages [26].

For this reason, a low-cost smart meter was developed that caters to the interests of the current energy sector and is capable of storing and sending data from the electrical system to a central system in real time. After being tested, the equipment achieved results that are very similar to those of products currently available on the market.

## 2. Materials and Methods

On the basis of the above assumptions, we designed an intelligent meter capable of picking up essential signals from the electric grid. The flowchart in Figure 1 displays the main components of the smart meter and their connections.

In Section 2.1 and Section 2.2, there is an explanation of the circuits used to convert voltage and current from the electric network to a voltage level that can be analyzed by the ESP32 microcontroller but similar to other conditioning circuits used with other microcontrollers [27]. The zero-crossing circuits of voltage and current sine waves are shown in Section 2.3, and these are responsible for calculating various parameters, such as frequency and the angles between the phases. In addition, in Section 2.4, all the points of the microcontroller used are elucidated, with a particular focus on its 12-bit ADC, which allows the reading of fundamental signals with a good degree of precision.

### 2.1. Voltage Conditioning Circuit

The voltage conditioning circuit conducts and converts the voltage measured directly in the electric network to the microcontroller’s input. In Figure 2, the blocks of the circuit implemented can be seen, and Figure 3 shows the schematic circuit diagram, where the collected voltage is reduced through the arrangement of resistors, which can be regarded as a significant advantage because of the cost and the fact that its linearity is obtained with the circuit. The V_INPUT component represents the connector that receives the voltage from the power grid, and the labels VA, VB, and VC are the processed signals that the microcontroller receives for reading. The block, called the ‘filter’, consists of capacitors, which have the function of filtering any noise from the electric network and decoupling. Moreover, the voltage reaper block, formed of diodes, generally limits the voltage values and thus protects the circuit’s other components.

In the arrangement of resistors, there is a common node called ‘ref’, which is directly connected to the buffer circuit (Figure 4). This circuit is a resistive divider designed to generate the reference voltage with an operational amplifier in the buffer configuration so that the isolation between the circuits is established. This circuit’s primary function is to insert a DC component into the wave so that the minimum value is 0.3 V and the maximum value is 1.2 V, which causes this wave to occur in the range of the best linearity of the 12-bit ADC of the ESP32 as shown in Section 2.4.

All the circuits were simulated in LTSpice XVII v17.0.27.0 [28] software to validate the proposed circuit properly. As a way of simulating the circuit shown in Figure 2, the input voltage was 311 Vpp or 220 Vrms and emitted an output signal between 0.15 V and 1.15 V. When sizing the circuit, it is also essential to measure the power above each resistor in the array, and by carrying out simulations, a maximum power of 150 mW was obtained.

### 2.2. Current Conditioning Circuit

The current conditioning circuit (Figure 5) follows the same principle as the voltage conditioning circuit; its main difference is in how the signal represents the input from the current transformers and injects a current wave with a maximum amplitude of 5 A. In Figure 6, the schematic of the circuit (the I_INPUT component) represents the connector that receives the current from the current transformers, and the labels IR, IS, and IT are the processed signals that the microcontroller receives for reading. The input current passes through a circuit that will convert it into voltage and leave it at the appropriate thresholds as in the conditioning circuit of tension. The resulting current wave has an amplitude of 0.7 V, which is also inserted into a DC component, and leaves the wave between 0.3 V and 1 V. The other blocks in Figure 5 have the same operation as those in Figure 2.

### 2.3. Voltage and Current Zero-Crossing Circuit

There is a need to implement a zero-crossing circuit to read parameters such as the frequency and angles of each voltage and current phase. The main purpose of the zero-crossing circuit is to detect the passage of the sine wave through the zero volts line, or reference voltage. This information is essential to know exactly when the sinusoid passed through zero and thus make the necessary comparisons and mathematical operations to calculate the desired parameters.

The proposed circuit relies on a comparator circuit for both voltage and current applications. This circuit makes use of an operational amplifier as a comparator, and takes the values coming from the voltage and current reference circuits as a benchmark. The signal that has to be compared is the sinusoidal signal of each voltage and current phase after the conditioning circuit.

### 2.4. Microcontroller ESP32

The ESP32 chip (Figure 7) refers to a microcontroller chip produced by Espressif, which has for some time been regarded as one of the most powerful and renowned microcontrollers on the market; its most powerful features include its processing speed, accessibility, and connectivity, the last of which should be highlighted because of its intelligibility through its Wi-Fi and Bluetooth connection.

The ESP32 was designed with a model that can be either single or have dual-core 32-bits, with two physical processing cores, that can work with clock frequencies of up to 240 (MHz). In addition, it has the huge advantage of possessing a storage capacity that is exponentially more significant than that of the already established Arduino development boards, which has twice as much flash memory as the ATmega 2560 model [29].

One of the crucial features for selecting this chip ESP32-WROOM-32DC is its ADC, which integrates two 12-bit SAR ADCs and supports measurements on 18 channels (analog-enabled pins). The (ULP) co-processor in ESP32 is also designed to measure voltage while operating in sleep mode, which allows low-power consumption. The CPU can be woken up by a threshold setting and/or by other triggers. The ADCs can be configured with appropriate settings to measure the voltage on 18 pins maximally [30].

The ADC resolution can map the input voltage from 0 to 3.3 Vdc depending on the resolution setting (9–12 bits) and the maximum voltage reading configured by the attenuation filter and the recommended ranges for each voltage (Table 1).

Espressif noted that, by default, there is a difference of ±6% in the measured results between chips. This difference takes into account the region of the ADC that has no sensitivity, below 0.14 V, and the region of nonlinearity, above 2.5 V. However, the hardware was developed to operate in the linearity region of the ADC, between the range of 0.14 V and 2.5 V. Even with a low error, as with any equipment that will be launched to the market, it is essential to calibrate the microcontrollers, and one of the guidelines is to select a single reference value for the project [31].

## 3. Prototyping

The EasyEDA STD v6.5.34 software was chosen as a tool for the development of the project, since it allows the creation of online schemes, and facilitates remote collaboration. The software offers suitable features for public and private sharing, along with standard tools for 2D and 3D visualization. In addition, it allows you to send the project to manufacture within the system at any time, easily and quickly.

Figure 8 displays the 3D visualization of the project generated in EasyEda.

Table 2 contains all the components and their respective prices, which were obtained from the market in small-scale purchases. It is worth noting that these costs may vary depending on the location and amount purchased.

A survey was carried out of all the components needed to manufacture a printed circuit board (PCB), and Table 2 is set up with its respective components, amounts, and prices. As shown in the table, the developed hardware has a complete set of resources to measure and monitor key electrical parameters, such as voltage, current, power, and the power factor with a lower market price, compared with other available products, which allows consumers to access reliable, high-quality monitoring tools without exceeding their budget. This option is particularly attractive for companies, industries, and residential users who want to control and optimize energy consumption, ensure energy efficiency and make a cost reduction.

The power input stages are an essential part of a meter’s circuit, and thus require special care when defining the thickness of their tracks.

The IPC-2221A [32] is the foundation design standard for all of the IPC-2221A design series and lays down the generic requirements for the design of PCBs and other means of assembling the components or interconnecting structures. This should be taken into account to ensure the correct sizing of the required track width for a maximum current of 5 A, which is found in the power stage of the current conditioning circuit. To avoid overheating the track and any possible collapse of the track on the board caused by overheating, a track of 54.437 mil was used in accordance with the calculated values and this was in accordance with the IPC standard Equation (Equation 1):(1)$$\begin{array}{c} A={\left(\frac{I}{k\times \Delta T^{0.44}}\right)}^{\frac{1}{0.725}} \end{array}$$
where:

*k* is the coefficient;

$k_{e}=0.048$ (for external layers);

$k_{i}=0.024$ (for inner layers);

*A* is the cross-section area (square mills);

*I* is the current (amps);

ΔT is the change in the temperature above ambient (°C).
$$I=5A\;K=0.048\;\Delta T=10\;{}^{\circ}C\;A={\left(\frac{5}{0.048\times 10^{0.44}}\right)}^{\frac{1}{0.725}}\;\Rightarrow \;A=150.028$$

With a chosen thickness of A mils, the trace width W mils can be calculated (Equation 2):(2)$$\begin{array}{c} W=\frac{A}{1.378\times E} \end{array}$$
where:

*A* is the cross section area in mils;

*W* is the trace width;

*E* is copper weight.


$$E=2oZ/ft^{2}\;Width=\left(\frac{150.028}{1.378\times 2}\right)\;\Rightarrow \;Width=54.437\;\mathrm{mil}$$


We manufactured the remaining traces on the board with a thickness of 19.685 units of measurement, which is equal to one thousandth of an inch (mil) for low-current signals.

In the power stage, a parallel combination of resistors was used for current inputs and a series combination of resistors for voltage inputs. We used resistor combinations instead of a single higher-power resistor to ensure there was a larger heat dissipation area for these components. Additionally, the use of standard resistors on the market ensures greater availability and a significant cost reduction in the power stage.

Voltage follower operational amplifiers were used on the reference signal of the voltage dividers to ensure greatly improved stability in the ADC readings.

The Hilink model switching power supply created a compact and easy-to-install meter. This power supply is cost effective, provides sufficient power, has a compact size, and is highly reliable.

## 4. Calibration and Standardization

When considering the use and development of measurement tools, one of the main points to be observed is the reliability of the data obtained; in view of this, methods must be employed that attest to the quality of the data obtained by the smart meter. The method used to perform the calibration was divided into two stages, namely, analysis and generation of adjustment curves based on known loads in a controlled environment, as well as real loads over a long period of time. The need for this method is well known and arises because the ADC input that is used can be found in the ESP32, which, like any component in the real world, has a certain error curve that can be reduced through the mathematical technique of polynomial regression.

### 4.1. Calibration Platform

In order to calibrate the ESP32 ADC input, it is necessary to assemble a resistive circuit in the laboratory, and this has the following components:A three-phase voltage regulator;Cables;Equivalent resistors of 33.33 ohms, with a power of 900 watts per phase;Three 5 amp output current transformers;Two secure digital cards;Multimeter;CW500 energy analyzer.

For this reason, as shown in Figure 9, there is a basic diagram that was set up in the laboratory. The purpose of this diagram is to collect data from a known and approximately linear load, consisting of resistors used in alternating voltage networks. The meters were connected to the system to obtain the data as illustrated in Figure 10. Both the CW500 and the smart meter saved the data on their respective SD cards, which were used to create the fitting equation as explained in Section 4.2.

### 4.2. A Performance Calibration Using the Intelligent Meter

The calibration of the smart meter was performed using CW500, from the company Yokogawa T&M (Test&Measurement), which is a company specialized in test and measurement equipment. The CW500 is a portable electrical power quality analyzer used for conducting a power quality analysis and an analysis in an electrical system. This device is capable of monitoring and recording electrical parameters, such as voltage, current, power, power factor, harmonics, transients and other irregularities in power quality [33], which means it is capable of the following:Detecting power quality problems;Evaluating energy efficiency;Diagnosing faults;Analyzing the performance of the electrical system.

On the basis of this information, the operation and maintenance of electrical systems can be optimized to ensure compliance with the regulatory standards and improve energy efficiency.

The calibration procedure begins with recording the acquired measurements from the CW500 meters, which record data every second, and the smart meter, which records data every three seconds. Next, we proceed to adjust the three-phase voltage regulator so that the voltage and/or current can be gradually varied, which allows a complete exploration of the voltage and/or current range. As a result, a discrete time series of the current and voltage present in the circuit (as shown in Figure 9) is obtained, and the information is collected by both meters. However, there is no guarantee that both meters will start measurements simultaneously or that their clocks are perfectly synchronized, and the time interval between the measurements may not be uniform. Thus, it is necessary to perform synchronization corrections on the obtained time series.

After the laboratory work is completed, the corrections are made a posteriori with data processing and software analysis conducted in Python (a high-level programming language) because of its clear and concise syntax as well as its wide range of data analysis tools.

The data obtained from the SD cards of each meter show a temporal deviation, that is, a differentiated number of samples. In the case of the current, there is a different scale because the current transformer in the circuit that connects the CW500 meter was used, while the smart meter was directly connected to the test bench. This is equivalent to a current transformer with a unitary transformation ratio. Owing to its constructive features, the current conditioning circuit in the smart meter is limited to a current of 5 amperes, a fact that does not affect the results obtained from the experiment. As a result of the identification of the voltage and/or current peak previously incorporated to facilitate the detection of the beginning of the tests, the corrected time series had an adjusted number of samples based on the repetition of samples from the smaller time series, and the scale was adjusted through a simple division.

On the basis of these data, when properly synchronized, the regression equation coefficients were generated for each range and variable and resulted in a precise mathematical representation of the relationship between the voltage and current. We return to the following polynomial Equations (Equation 3)–(Equation 8):

Equation for currents (A, B and C)
(3)$$\begin{array}{c} I_{A}=-3.1364973\times x^{2}+2.37540397\times x-0.15488245 \end{array}$$
(4)$$\begin{array}{c} I_{B}=-3.94758478\times x^{2}+2.71596739\times x-0.19799091\; \end{array}$$
(5)$$\begin{array}{c} I_{C}=-5.03076935\times x^{2}+3.08317143\times x-0.22882746 \end{array}$$

Equation for voltages (A, B and C):(6)$$\begin{array}{c} V_{A}=-3.73087704\times x^{2}+1.01952000\times x-2.31686713\times 10^{-5} \end{array}$$
(7)$$\begin{array}{c} V_{B}=-2.97068991\times x^{2}+1.04631432\times x-1.56420981\times 10^{-4} \end{array}$$
(8)$$\begin{array}{c} V_{C}=-1.84121196\times x^{2}+1.06992181\times x-2.62561829\times 10^{-4} \end{array}$$

A quadratic relationship was obtained because it was the fitting that best suited the data set. The reason for this is that, even though at first the data seemed to have a linear feature, it was possible to observe small fluctuations and even a certain curvature in the trend. These variations can affect the performance of the smart meter.

As the ADC calibration stage is crucial for the development of the entire project, a more precise and smooth adjustment was preferred. The fact that this adjustment was suitable can be illustrated in Figure 11, and the ’good fit’ is demonstrated through the error shown in Figure 12, where the error of the samples tends to zero.

With the aid of the previously collected voltage and current data, the method for plotting curves was employed to analyze the reliability of this data, using an error approach based on the absolute difference metric.

We can observe in Figure 12 that the bench calibration process showed effectiveness as expected. We noticed that the curves obtained through the calculated polynomial fitting equations, when compared with the measurements originally obtained by the CW500, showed a low absolute error (error curves named “predicted”), close to zero. For a better understanding of the quality of this fit, it is possible to compare, in the same Figure 12, the absolute error between the CW500 and the values measured by the smart meter before using the polynomial fitting equations (error curves named “smart meter”).

Another point that attests to the quality of this fit is the comparison between the curves obtained through the polynomial equations and the CW500; the current measurements, which have greater variability than the voltage measurements, exhibit a low absolute error, with an average around zero. It is important to emphasize that the presence of some error in the meter comparison, even after the adjustments, does not indicate a design flaw but rather a constructive characteristic of the worked hardware. The practical results of using these adjustments are discussed in more detail in Section 5.

## 5. Results and Discussion

### 5.1. System Analysis in Practice

After the completion of the bench calibration, the CW500 meter was used. This is an important tool for checking the reliability of the obtained data. This meter is recognized for its precision and ability to monitor anomalies in the power system, including voltage and current fluctuations, and record up to the 50th order of harmonics, in both single-phase and three-phase systems.

The advanced interface of the CW500, including a graphical display and options for data collection through Bluetooth and SD cards, makes the equipment easy and intuitive to use. In addition, the data analysis software encompasses visualization, storage, and analysis of the data acquired by the CW500. As a portable energy meter and analyzer, the CW500 uses current clamp probes and alligator clips to perform its measurements as described in the user manual provided by the company [33]; all of these features make the CW500 a versatile tool for the development of low-cost meters.

As a result, it was possible to capture these data in a similar way to what was carried out in the bench calibration but this time over a longer period (from 31 December 2021 to 12 January 2022), which is approximately fourteen consecutive days. We were able to export these cataloged data into comma-separated value files, which was undertaken both with the CW500 and the smart meter. Afterwards, these data were subjected to event synchronization with the aid of software implemented in Python.

#### 5.1.1. Results and Field Test Analysis

In order to analyze and evaluate the performance and accuracy of the smart meter, and thus determine its suitability for similar applications, a comparative analysis was conducted between the CW500 devices and the smart meter. This analysis was based on voltage and current measurements taken over different time periods. One of these analyses was conducted with data sets collected over a one-day period.

On the basis of Figure 13, it can be seen that the voltages had excellent performance, with the data captured by the smart meter being very similar to those obtained from the CW500 meter. In our analyses of the voltages, we found relative errors of around 0.5%, except for the very occasional occurrence of outliers. However, when we analyzed the relative errors found in the currents, there were values that exceeded 100% with a recurrence of about a 50% relative error rate.

With regard to electric currents, it should be noted that discontinuities were detected in the values collected by the smart meter, which resulted from the lack of precise synchronization between this device and the CW500 meter. This factor led to a relatively high error rate since the error rates linked to the currents do not accurately reflect the correspondence between the average value of the time series but only the RMS value at the time of measurement.

#### 5.1.2. Power, Power Factor, Frequency, and Harmonic Results in Field Test

Excellent performance was achieved in the measurement of frequencies. To obtain these data at the implemented hardware level, we used the methods described in Section 2.3, and these were then subjected to a field test to assess the reliability.

In Figure 14, it is noted that the frequency remained at 60 Hz, which is the current standard frequency in Brazil, where all the field tests were conducted. In our calculations of relative error, there was an error rate of 0%, except for outliers that may have appeared because of connection problems during the data-acquisition stage.

On the basis of the data on frequency, voltage, and current that were collected (as shown above), it was possible to determine the total harmonic distortion rates, as illustrated in Figure 15. In addition, we carried out measurements of the active power, which are represented in Figure 16. From these measurements, it was also possible to determine the PF, and the results are shown in Figure 17; these demonstrate the wide range of possible applications with the smart meter, especially in view of its low cost.

A similarity can be found between the values read by the CW500 meter and the developed smart meter. However, some values deviated from what was expected, owing to the accumulated error of the base samples, and in this specific case, it showed a relative error in the current. But as explained earlier, it is not possible to determine whether or not this affects the reliability of the constructed device.

The fact that the smart meter provides measurements of harmonics up to the 50th order, at a low cost, is a big differential. The presence of harmonics at lower frequencies can affect the correct operation of electrical and electronic equipment, such as motors, transformers, power supplies, and lighting devices. In addition, it is important to ensure the quality of the electrical energy supplied to consumers and to check that the electricity meters are measuring the amount of electricity consumed correctly. It should be noted that higher-frequency harmonics, above the 50th harmonic, can be more easily filtered and thus are less at risk in the measurement of electrical energy.

The smart meter also has the ability to measure the angle between the phases of the electric grid. This is due to the zero-crossing detection circuit described in Section 2.3. The obtained results are promising, as we were able to measure real-time phase angles, and make a further expansion of the range of applications for the meter at a low cost. In an increasingly complex electrical system with distributed network systems, the constant monitoring of phase angles [34,35] is necessary to prevent imbalances and distortions in the grid, which can make it harder to protect the electrical system.

### 5.2. Systems Analysis Theory

As well as the CW500, the equipment used to validate our system, other devices for analyzing energy quality were examined to determine which parameters should be selected for acquisition and how the data analysis would be conducted. A benchmark was established for comparing the smart meter with the Fluke 1770 series three-phase power quality analyzer, which is a reference product in the power quality measurement equipment market.

This tool is able to carry out the measurement of electrical parameters, such as voltage, current, frequency, active and reactive power, power factor, and energy, among other factors, and thus, it allowed a detailed analysis of the electrical performance of the system. It possesses advanced features to analyze the power quality and can detect and record harmonic distortion, transients, short duration interruptions, voltage spikes, frequency variations, load imbalance and other common problems related to power quality. Its functions include the real-time monitoring of electrical parameters and power quality, which enables it to detect immediate problems and take corrective measures more quickly and efficiently. The analyzer has an extensive data-logging capability, which allows measurement data to be stored internally for later analysis. This is useful for discovering trends, failure patterns, and recurring problems over time. The instrument also features an intuitive, easy-to-use user interface that has a quick setup and can efficiently navigate the analyzer’s menus and functions, while including analysis software to complement the Fluke 1770. These tools allow an in-depth analysis to be conducted of the registered data, as well as the compilation of personalized reports and the graphical visualization of the results.

Since it is believed to be one of the most complete and reliable devices on the market, its cost is quite high, currently around $7739.99 (US dollars). For this reason, the most relevant parameters were initially evaluated for our product, on the basis of which a table was prepared, (Table 3) for drawing a comparison between the smart meter, CW500 and the Fluke 1770 series three-phase power quality analyzer, where the “X” symbol indicates that the equipment has the feature and the symbol “-” indicating that it does not have.

## 6. Conclusions and Suggestions for Future Work

### 6.1. System Analysis in Practice

Although it depends on a low budget, the analysis conducted in this article shows that the technology embedded in the smart meter can ensure a certain degree of reliability and accuracy. This makes its implementation possible, as it can offer several advantages to energy suppliers and consumers, such as the accurate real-time measurement and remote monitoring of power quality data (including voltage and current parameters, active power, power factor, harmonic distortion, and phase angles). This raises consumer awareness and allows energy providers to upgrade power grids promptly. In addition, replacing commercial meters with smart meters can lead to long-term cost savings, even if a higher initial investment is required.

As seen in Section 5, the data of the smart meter proved to be reliable since their relative error in the frequency measurement was 0%. Despite the difficulties over synchronization we found, there was an error of approximately 0.5% in the voltage measurement and we achieved consistent measurements of the current and power factor over time. In view of this, in light of the parameters required for energy quality measurement, the smart meter proved to be a low-cost and reliable option.

The approximate final cost of the meter is $82.77 (as shown in Table 2 of Section 3). However, if the (CT) is removed, the final cost would be reduced to $34.77, a price much lower than any quality energy meter currently available on the market. This price difference provides a more affordable choice for those seeking to purchase a high-quality energy meter without compromising the performance or any necessary features.

When the value of the meter described in this article is compared with a commercial model like the ISSO-DMI P500Rv2 (which has an approximate price of $996.00), it can be seen that the described meter is about 10 times cheaper.

It is also possible to envisage future implementations in the field, such as the use of embedded artificial intelligence in smart meters with the aim of preventing failures, detecting devices, and preventing fraud in the electric grid. With technological advances, it is expected that these solutions will become increasingly accessible and widely used, and will provide benefits not only to electricity companies but also to consumers.

### 6.2. System Analysis Theory

From Table 3, it is possible to show that the main parameters to determine the quality of energy were selected for the smart meter to acquire. As a result, we can benefit from the fact that the data collection and analysis can be carried out remotely, that is, once the smart meter is installed, it will not be necessary to handle it on the distribution board, except when a problem has been discovered. Additionally, the hardware can be left running continuously to monitor the power quality over time. This will allow patterns and trends to be determined, as well as the detection of recurring or new problems, which will greatly benefit companies and factories that employ professionals specialized in correcting these failures. However, companies that do not have these professionals end up at a disadvantage both in terms of having to install the system and correct problems, as well as tracing patterns in their energy distribution since specialist knowledge is required for this.

## Figures and Tables

**Figure 1 sensors-23-07210-f001:**
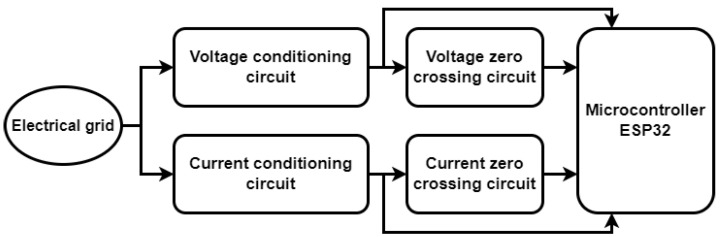
Flowchart of smart meter.

**Figure 2 sensors-23-07210-f002:**
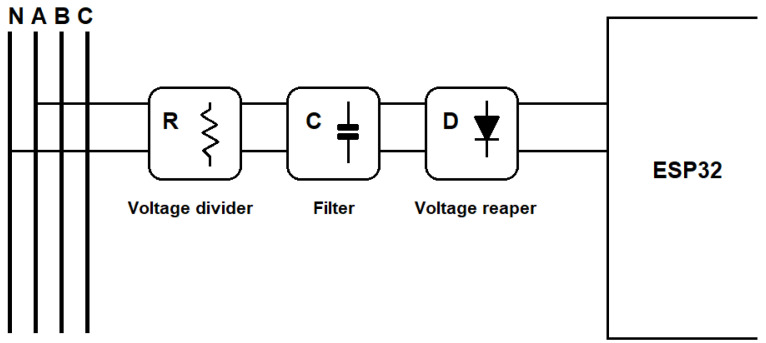
Conditioning circuit of voltage.

**Figure 3 sensors-23-07210-f003:**
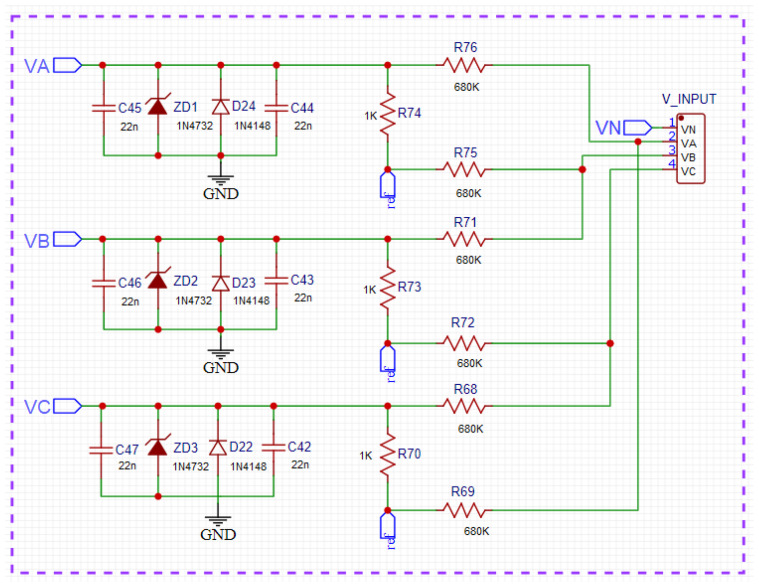
Schematic diagram of the voltage conditioning circuit.

**Figure 4 sensors-23-07210-f004:**
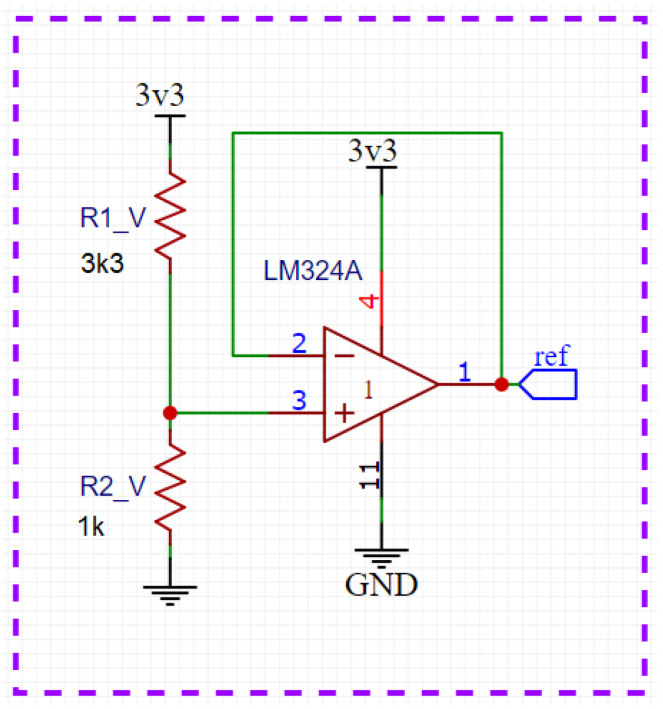
Circuit of reference voltage.

**Figure 5 sensors-23-07210-f005:**
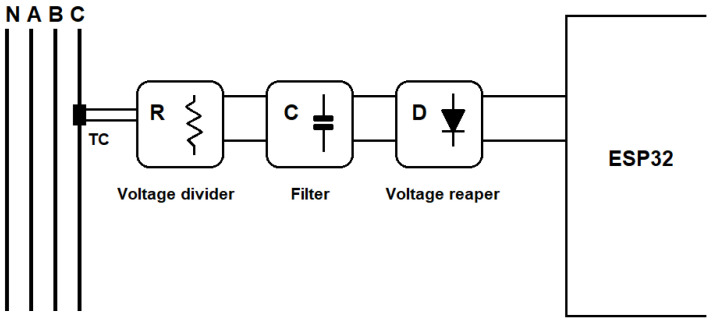
Current conditioning circuit.

**Figure 6 sensors-23-07210-f006:**
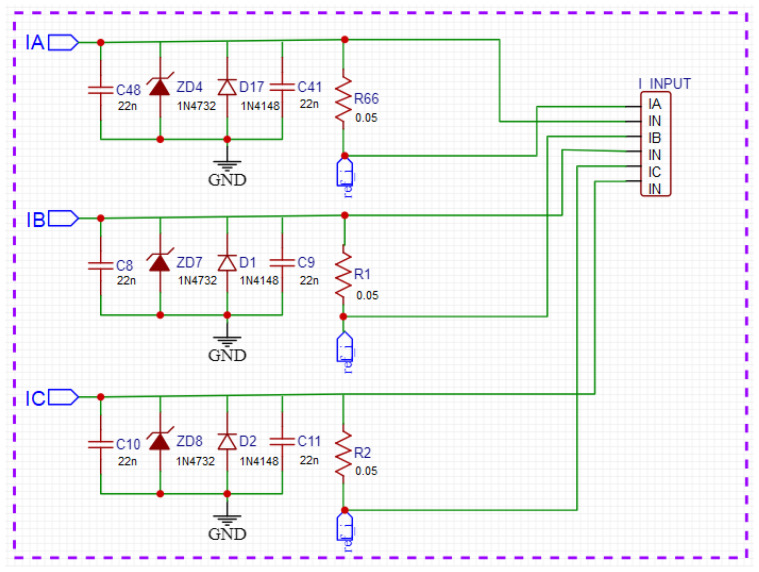
Schematic diagram of the current conditioning circuit.

**Figure 7 sensors-23-07210-f007:**
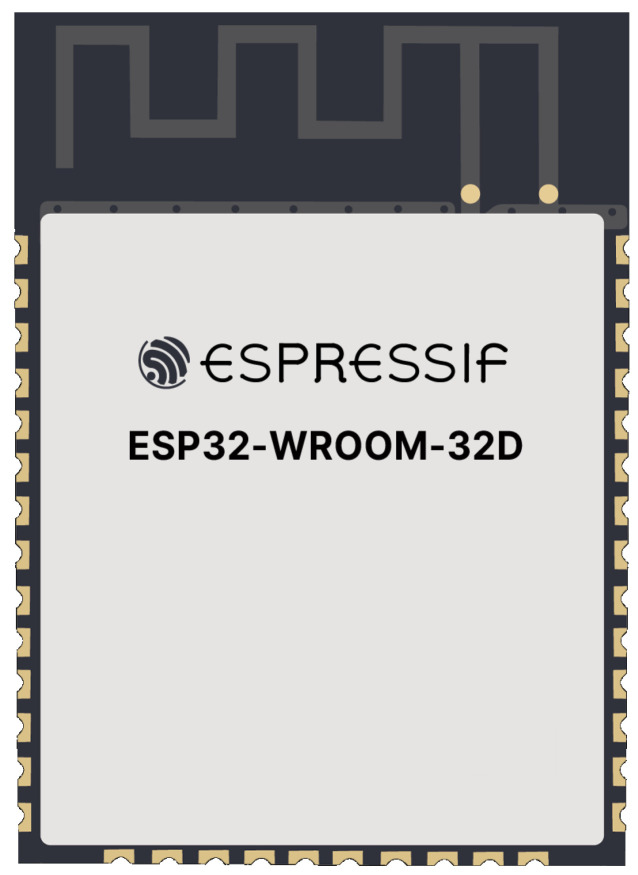
Chip ESP32.

**Figure 8 sensors-23-07210-f008:**
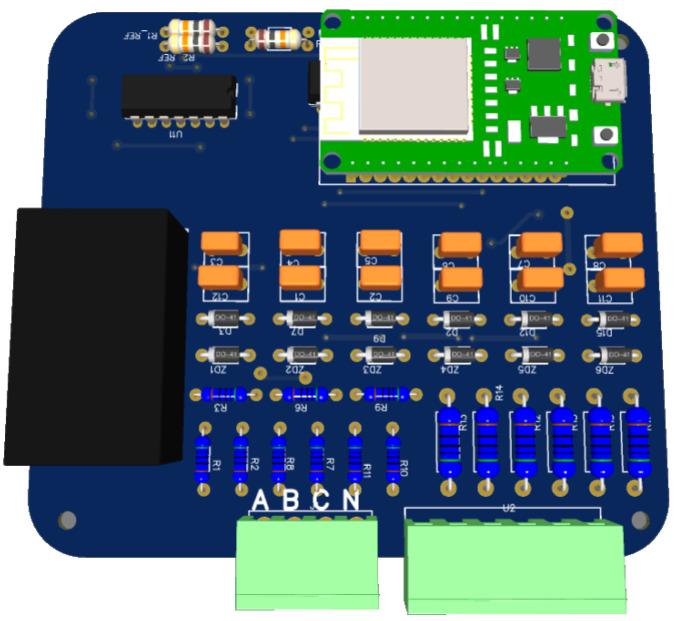
View in 3D.

**Figure 9 sensors-23-07210-f009:**
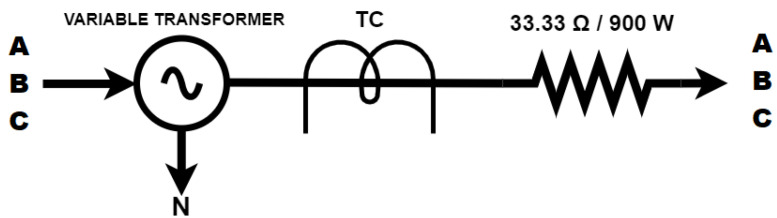
Circuit diagram of the laboratory setup.

**Figure 10 sensors-23-07210-f010:**
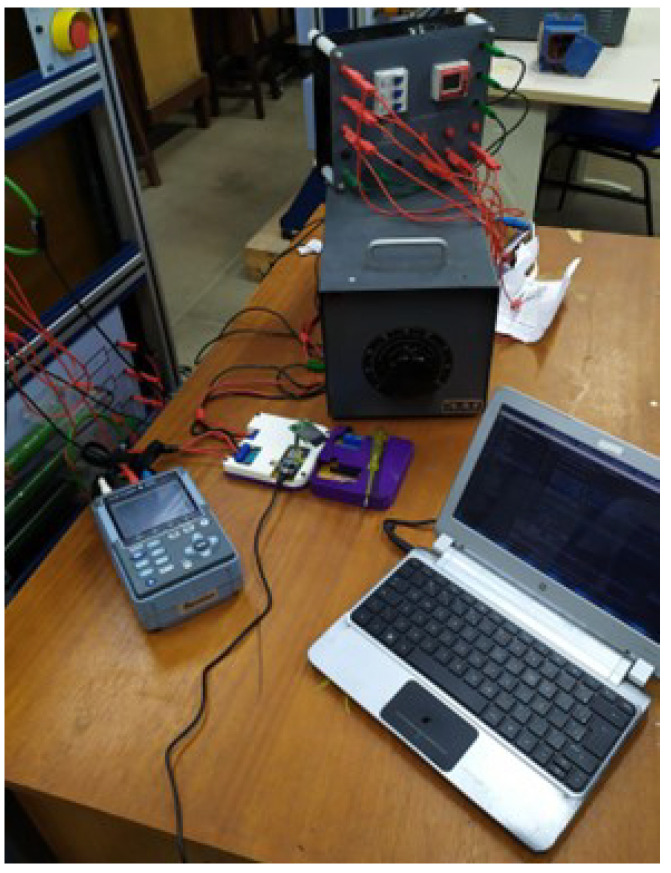
Bench-mounted circuit with the smart meter offline and the CW500 in parallel.

**Figure 11 sensors-23-07210-f011:**
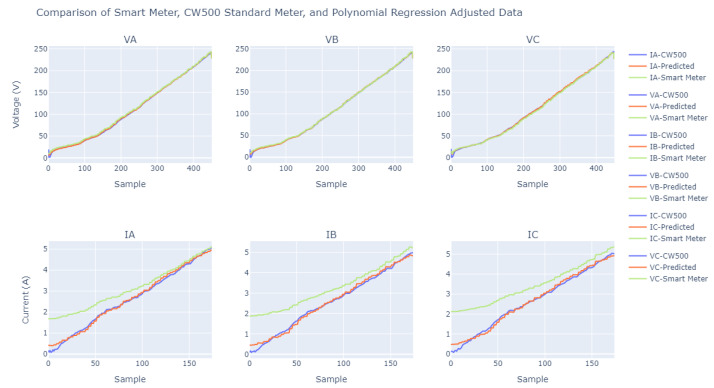
Comparative analysis of curves: results before and after calibration.

**Figure 12 sensors-23-07210-f012:**
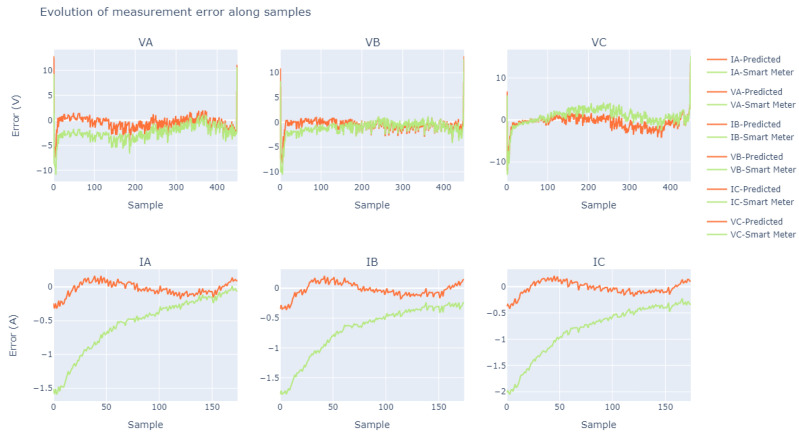
Graph illustrating the evolution of measurement error over time.

**Figure 13 sensors-23-07210-f013:**
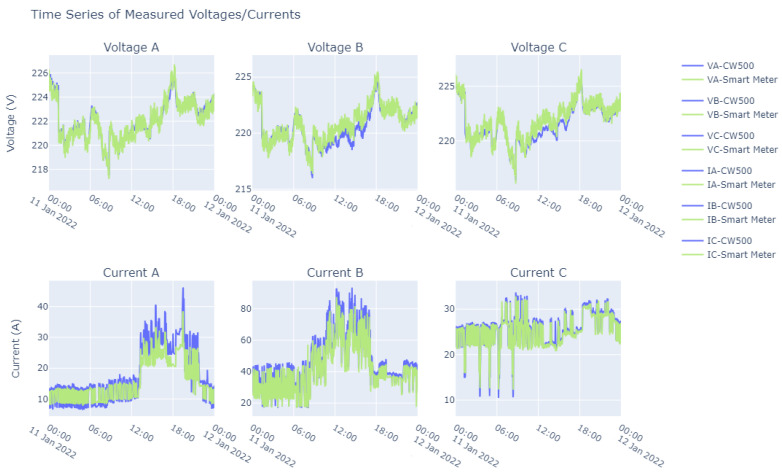
A one-day comparative analysis between CW500 and a smart meter.

**Figure 14 sensors-23-07210-f014:**
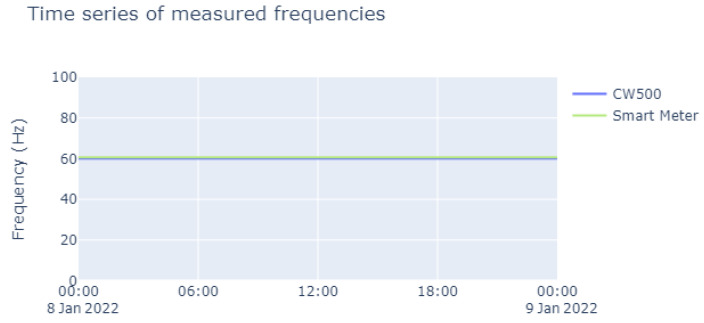
Time series of measured frequencies over the course of a day.

**Figure 15 sensors-23-07210-f015:**
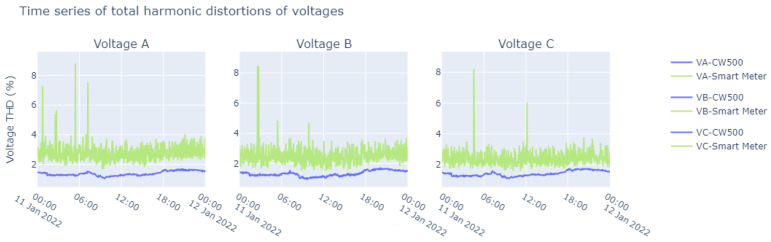
Time series of total harmonic distortions of voltages during the course of a day.

**Figure 16 sensors-23-07210-f016:**
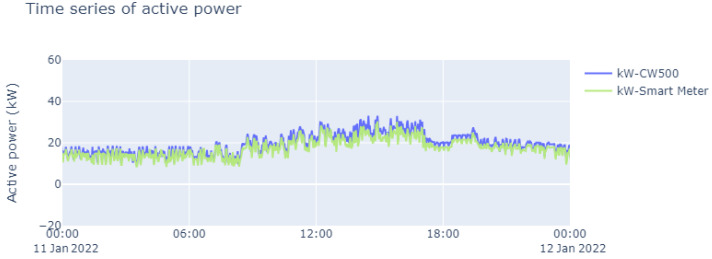
Time series of active power during the course of a single day.

**Figure 17 sensors-23-07210-f017:**
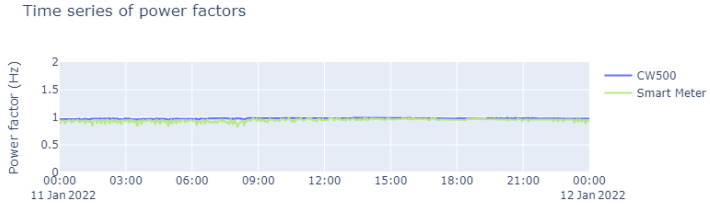
Time series of power factors during the course of a day.

**Table 1 sensors-23-07210-t001:** Table with the recommended attenuation filters for each range.

Attenuation	Voltage Range
0 dB	100–950 mV
2.5 dB	100–150 mV
6 dB	150–1750 mV
11 dB	150–2450 mV

**Table 2 sensors-23-07210-t002:** Table of prices.

Components	Amount	Dollar	Sum
AMS1117-3.3	1	$0.30	$0.30
Resistor 10k 1/8 W	3	$0.04	$0.12
Resistor 680k 1/8 W	6	$0.04	$0.24
Resistor 1k 1/8 W	5	$0.04	$0.20
Diode 1N4732A	6	$0.03	$0.18
Resistor 2Ω	12	$0.17	$2.04
Capacitor 22 uF	3	$0.05	$0.75
Capacitor 22 nF	6	$0.05	$0.30
Capacitor 100 nF	10	$0.06	$0.60
Diode 1N4148	6	$0.03	$0.18
Resitor 3k3	5	$0.04	$0.20
LEDS	3	$0.06	$0.18
Pin Header Socket	1	$0.80	$0.80
RST	1	$0.14	$0.14
ESP32-WROOM-32DC	1	$12.00	$12.00
LM324A-CORRENTE	1	$0.23	$0.23
LM324A-TENSAO	1	$0.23	$0.23
Capacitor 100 uF	1	$0.18	$0.18
HLK-PM01 5 v 600 mA	1	$5.80	$5.80
PCB	1	$1.60	$1.60
CASE 3D	1	$5.00	$5.00
Conn 2EDGRC-5.0 M 6P 90${}^{\circ }$	1	$0.73	$0.73
Conn 2EDGK-5.0 F 6P	1	$1.18	$1.18
Conn 2EDGRC-5.0 M 4P 90${}^{\circ }$	1	$0.53	$0.53
Conn 2EDGK-5.0 F4P	1	$1.06	$1.06
Current Transformer	3	$16.00	$48.00
	Total		$82.77

**Table 3 sensors-23-07210-t003:** Comparative diagram of the equipment being analyzed for power quality.

Features	Fluke 1770 Series	CW500	Smart Meter
Measuring power and power quality parameters	X	X	X
Power directly from measurement circuit	X	-	X
Data acquisition offline	X	X	X
Data acquisition online	X	-	X
High-speed voltage transient capture	X	-	-
Display for viewing data	X	X	-

## Data Availability

Not applicable.

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
