# Peer review of "Development a Low-Cost Wireless Smart Meter with Power Quality Measurement for Smart Grid Applications"

_sensors, 2023, doi:10.3390/s23167210_

Round 1

Reviewer 1 Report

The topic is interesting and it is adapt to this journal. There are few comments and/or suggestions to improve the manuscript.

-Clarify better the innovative novelty of this work in the abstract.

-Read articles to understand the structure of Sensors. The following structure would be preferable based on the Microsoft Word template file: 1. Introduction (1.1, 1.2, 1.3.), 2. Materials and Methods (2.1, 2.2., 2.3.), 3. Results (3.1, 3.2, 3.3), 4. Discussion (4.1, 4.2, 4.3), 5. Conclusions. The discussion chapter is missing!

-The Introduction chapter is basically a literature review. For this reason, each new paragraph must include a reference. This is missing in several paragraphs and should be corrected (E.g. 24-50).

-The advantages and disadvantages of the proposed method need to be better compared with other references/practical solutions.

-How can the presented method be easily used in practice? This needs to be explained in depth.

-Please search references to the equations. Equations should always be accurately and clearly referenced. 

-At the end of the study need to create a nomenclature / abbreviation table with units.

-

Reviewer 2 Report

This a good paper about Smart metering. It needs more details to be clear for the reader. Details circuit layout for each block are in Figures 2 and 3. Explanation of the circuits beneath these figures has no meaning without a circuit layout for each block. Moreover, some corrections to enhance the paper quality as:

1-      Repetition of text in the introduction section, as the same wording for power quality is mentioned in more than one paragraph in this section.  

2-      Some text typos as in Line 169 blocks instead of blocs.

3- The overall English of the paper has to be revised.

The overall English of the paper has to be revised (minor editing required). 

Reviewer 3 Report

Proposed paper is interesting, and well balanced. Paper requires minor corrections in sections 0-4, but as a result of authors’ simplification of text in mentioned sections the review of experiment outcomes (section 5) is for me impossible today. It is a major issue of proposed paper.

Abstract:

“In increasingly complex and interconnected electrical systems, this device is essential in various applications, such as smart grids, enabling real-time energy usage monitoring.” --> please consider replacement of is with will be:  In increasingly complex and interconnected electrical systems, this device will be essential in various applications, such as smart grids, enabling real-time energy usage monitoring.

2. Development:

Line 164 “And in the 2.4 section, all points of the microcontroller used will be elucidated, focusing mainly on its 12-bit ADC, which allows the reading of fundamental signals with good precision.” -> Please define precision in numbers. Because : Line 228 “Espressif noted that, by default, there is a difference of _6% in measured results between chips.”  This shows that good precision is 6%. It is popular knowledge that integrated adc in esp is not of perfect “quality” - so please clarify your thesis of quality in precise way. And give the board name with esp32 that was used in experiment. It is not clear what type and generation was used and discussed up to table 2 (ESP32-WROOM 32DC) that is ni page 8. That shows that requires some works on logical order of text have to be done

“Figure 2. Conditioning circuit of voltage” please give an electric scheme or change the title of figure.

In addition, the presentation of the electrical diagram of the entire analyzed sensor will significantly facilitate the understanding of the text of the publication. This is a simple task as authors write that “circuits were simulated in the LTSpice” so simply export blocks ow whole circuit. Then please mark places where voltages were measured and where were simulated.

Please apply the above recommendations to the content of sections 2.2 and 2.3

Please correct table 1 frequency was never measured in dB.

3 Prototyping

First paragraphs of prototyping section (I think up 278 line) have to be moved into introduction.

4. Calibration and Standardization

“Figure 6. Circuit diagram of the laboratory setup.” Is not correlated with equation s (3-8) so the reader has no idea how the author calibrated the transducer.

Therefore, the content of Fig. 8 is a puzzle for the reader where x where vabc, Iabs and where the number of the sample in the tested system, the circuit of which he does not know.

5. Analysis on the Experiment Outcomes

The same remark as above applies to Figures 9 and 10 thus review of experiment outcomes is for me impossible. Please give more precise information. For example “sample” on fig axis require sampling time. Please let me know how the error is calculated, etc.

The text of the work sometimes requires clarification, e.g.: “In increasingly complex and interconnected electrical systems, this device is essential in various applications, such as smart grids, enabling real-time energy usage monitoring.” --> Please consider replacement of is with will be:  In increasingly complex and interconnected electrical systems, this device will be essential in various applications, such as smart grids, enabling real-time energy usage monitoring.

Reviewer 4 Report

The article is interesting, and the topic is relevant for Sensor mdpi. The paper noted that the obtained data were properly synchronized, and the regression equation coefficients were generated for each range and variable, resulting in a precise mathematical representation of the relationship between voltage and current. Why this type of polynomial equation was chosen? Were sensitivity and adequacy checked?

Add an explanation of all variables and abbreviations.

The article is interesting, and the topic is relevant for Sensor mdpi. The paper noted that the obtained data were properly synchronized, and the regression equation coefficients were generated for each range and variable, resulting in a precise mathematical representation of the relationship between voltage and current. Why this type of polynomial equation was chosen? Were sensitivity and adequacy checked?

Add an explanation of all variables and abbreviations.

Reviewer 5 Report

-What is the novelty of this paper? Why is it important (or not) to existing knowledge?

-Kindly, refer eqs.that are not yours.

-Improve the conclusion.

-Please provide the reference in the correct format.

Moderate editing of English language required

Round 2

Reviewer 1 Report

This revised manuscript has addressed all my previous concerns. Therefore, I recommend that this paper in its current form is acceptable to the journal.

-

Author Response

We would like to express our heartfelt gratitude to the anonymous reviewers for their invaluable comments and suggestions, which have played a pivotal role in enhancing the quality of our manuscript significantly. Their expert feedback and constructive criticism have enabled us to refine our work and make it more impactful. Furthermore, we extend our sincerest appreciation to the reviewers for recommending the publication of our manuscript, as their endorsement has reinforced the value of our research. Without their dedicated efforts and expertise, this accomplishment would not have been possible. Once again, we sincerely thank the anonymous reviewers for their invaluable contributions to our work.

Reviewer 3 Report

I think that without a full scheme of proposed “smart meter” the publication is a waste time for readers.

Despite the correction, I still do not know where the IA, IB, IC currents flow and where the UA, UB, UC voltages are measured. If this is classified information, please do not publish the article. If this is non-confidential information, please provide a diagram with appropriate markings.

Currents IA, IB, IC and voltages UA, UB, UC according to the text search engine are not referred to in the text of the work. So the question arises, what do the authors mean?

“Figure 11 shows that the bench calibration process was perfectly satisfactory, since, 439 although the measurements had a relatively low error rate, we were able to reduce it to 440 close to zero with the polynomial equations.” I can see quite a big difference between current predicted and metered – green and red characteristics. For me it does not look like “perfect satisfactory”.

On diagrams 3 and 5, please enter the values of the elements.

Some sentences can be converted into passive voice, for example: "Figure 11 shows that the bench calibration process was perfectly satisfactory, since, although the measurements had a relatively low error rate, we were able to reduce it to close to zero with the polynomial equations". In my opinion "Figure does not show" instead "information is visible on the figure".

Reviewer 5 Report

Accept in present form

Minor editing of English language required

Author Response

(The authors gave the same response as above.)
